# Immunosuppressive Activities of Novel PLA_2_ Inhibitors from *Xenorhabdus hominickii*, an Entomopathogenic Bacterium

**DOI:** 10.3390/insects11080505

**Published:** 2020-08-04

**Authors:** Md. Mahi Imam Mollah, Aman Dekebo, Yonggyun Kim

**Affiliations:** 1Department of Plant Medicals, College of Natural Sciences, Andong National University, Andong 36729, Korea; mahiimam@yahoo.com; 2Department of Applied Chemistry, Adama Science and Technology University, P.O. Box 1888 Adama, Ethiopia; amandekeb@gmail.com

**Keywords:** PLA_2_, eicosanoid, *Xenorhabdus hominickii*, insecticide, immunity

## Abstract

**Simple Summary:**

Insect immune responses defend fatal attacks from entomopathogens. A Gram-negative *Xenorhabdus hominickii* exhibits high entomopathogenicity against lepidopteran insects. During the pathogenic processes, the bacteria suppress host insect immune responses by inhibiting phospholipase A_2_ (PLA_2_) enzyme activity with the bacterial secondary metabolites. PLA_2_ catalyzes eicosanoid biosynthesis. Eicosanoids mediate both cellular and humoral immune responses against various insect pathogens. This study identified eight different PLA_2_ inhibitors from the bacterial culture broth. Butanol extract of the culture broth possessed high potency to inhibit PLA_2_ activity. Subsequent fractionations led to identification of eight different compounds. The synthetic compounds also showed PLA_2_ inhibition and insecticidal activities. Furthermore, the addition of the bacterial PLA_2_ inhibitors significantly enhance other bacterial pathogenicity, suggesting its potential to be applied for developing novel insecticides.

**Abstract:**

Eicosanoids mediate both cellular and humoral immune responses in insects. Phospholipase A_2_ (PLA_2_) catalyzes the first committed step in eicosanoid biosynthesis. It is a common pathogenic target of two entomopathogenic bacteria, *Xenorhabdus* and *Photorhabdus*. The objective of this study was to identify novel PLA_2_ inhibitors from *X. hominickii* and determine their immunosuppressive activities. To identify novel PLA_2_ inhibitors, stepwise fractionation of *X. hominickii* culture broth and subsequent enzyme assays were performed. Eight purified fractions of bacterial metabolites were obtained. Gas chromatography and mass spectrometry (GC-MS) analysis predicted that the main components in these eight fractions were 2-cyanobenzoic acid, dibutylamine, 2-ethyl 1-hexanol, phthalimide (PM), dioctyl terephthalate, docosane, bis (2-ethylhexyl) phthalate, and 3-ethoxy-4-methoxyphenol (EMP). Their synthetic compounds inhibited the activity of PLA_2_ in hemocytes of a lepidopteran insect, *Spodoptera exigua*, in a dose-dependent manner. They also showed significant inhibitory activities against immune responses such as prophenoloxidase activation and hemocytic nodulation of *S. exigua* larvae, with PM and EMP exhibiting the most potent inhibitory activities. These immunosuppressive activities were specific through PLA_2_ inhibition because an addition of arachidonic acid, a catalytic product of PLA_2_, significantly rescued such suppressed immune responses. The two most potent compounds (PM and EMP) showed significant insecticidal activities after oral administration. When the compounds were mixed with *Bacillus thuringiensis* (Bt), they markedly increased Bt pathogenicity. This study identified eight PLA_2_ inhibitors from bacterial metabolites of *X. hominickii* and demonstrated their potential as novel insecticides.

## 1. Introduction

Entomopathogens can invade insects through virulence factors while insects can defend against these pathogens through their immune systems. Insect immunity is innate. It consists of cellular and humoral responses [1]. Upon a pathogen infection, insects can recognize the invading pathogen based on pathogen-specific molecular patterns such as lipopolysaccharides, peptidoglycan, or β-1,3-glucan [2] and express pathogen-specific immune responses with the help of immune mediators such as biogenic monoamines, cytokines, and eicosanoids [3]. These immune mediators activate hemocytes to perform cellular immune responses such as phagocytosis, nodule formation, and encapsulation depending on invader type and amount [4]. They can also trigger antimicrobial peptide (AMP) production from fat body and other tissues [5] as well as prophenoloxidase (PPO) activation cascade [6] to achieve humoral immunity.

All three types of eicosanoids (prostaglandins, leukotrienes, and epoxyeicosatrienoic acid) in insects are known to mediate nonself recognition signals against various insect pathogens, including bacteria, fungi, viruses, and parasitoid eggs in insects [7]. Nonself recognition can stimulate eicosanoid biosynthesis by activating phospholipase A_2_ (PLA_2_) in insects [8,9]. Inhibition of PLA_2_ can prevent eicosanoid biosynthesis and suppress immune responses to microbial pathogens [10,11,12].

Two entomopathogenic bacteria, *Xenorhabdus* and *Photorhabdus*, are symbiotic to entomopathogenic nematode genera *Steinernema* and *Heterorhabditis*, respectively [13,14,15]. Infective juveniles (IJs) of host nematodes can enter the target insect’s hemocoel through natural openings such as mouth, anus, and spiracles [16] and then release their symbiotic bacteria from the nematode’s intestine. These released bacteria can suppress insect immune responses by inducing hemolysis, degrading AMPs, or suppressing PPO activation via inhibition of eicosanoid biosynthesis [17,18]. Under immunosuppressive conditions, bacteria can multiply and kill host insects. In the insect cadaver, IJs can grow and reproduce to form new generations. Subsequent IJs can re-associate with those symbiotic bacteria and come out of the insect cadaver to look for other target insects [13]. To inhibit host immune responses, these symbiotic bacteria can inhibit target insect’s PLA_2_ with their secondary metabolites [17,19]. At least eight different secondary metabolites have been identified from culture broth of *Xenorhabdus* and *Photorhabdus* [20,21]. 

*Xenorhabdus hominickii*, an entomopathogenic bacterium, is symbiotic to a nematode *Steinernema monticolum* [22]. Cultured broth of *X. hominickii* contains secondary metabolites that can inhibit the activity of PLA_2_ [23]. A secondary metabolite oxindole has been identified from an organic extract of the culture broth of *X. hominickii* [23], suggesting that *X. hominickii* might produce novel PLA_2_ inhibitors other than compounds identified in other bacteria. The objective of this study was to identify novel bacterial metabolites responsible for PLA_2_ inhibition from the bacterial culture broth of *X. hominickii*. To this end, *X. hominickii* culture broth was sequentially fractionated and analyzed for PLA_2_ inhibition. Purified compounds possessing PLA_2_-inhibitory activity were subjected to gas chromatography and mass spectrometry (GC-MS) analysis. Candidate PLA_2_ inhibitors were then analyzed for their inhibitory activities against cellular immune responses. Their insecticidal activities were also analyzed for the development of novel pesticides.

## 2. Materials and Methods

### 2.1. Insect Rearing and Bacterial Culture 

A laboratory colony of beet armyworm, *Spodoptera exigua*, was used in this study. Larvae were reared with an artificial diet [24] at 25 ± 1 °C with 16 h of light and 8 h of darkness. Adults were provided with a 10% sucrose solution. Fifth instar (L5) larvae were collected from cohorts for bioassays and hemolymph collection. *X. hominickii* ANU101 was cultured in Tryptic Soy Broth (TSB: Difco, Sparks, MD, USA) for 48 h at 28 °C with shaking at 180 rpm [22]. *Bacillus thuringiensis* subsp. *kurstaki* was obtained from Hanearl Science (Taebaek, Korea) and cultured in a nutrient broth medium (0.5% peptone and 0.3% beef extract) at 30 °C for 48 h. For endospore formation, cultured bacteria were further incubated at 4 °C for at least 24 h.

### 2.2. Chemicals

Arachidonic acid (AA: 5,8,11,14-eicosatetraenoic acid) was purchased from Sigma-Aldrich Korea (Seoul, Korea) and dissolved in dimethyl sulfoxide (DMSO). A PLA_2_ surrogate substrate, 1-hexadecanoyl-2-(1-pyrenedecanoyl)-*sn*-glycerol-3-phosphatidylcholine, was purchased from Molecular Probes (Eugene, OR, USA) and dissolved in high grade ethanol (Sigma-Aldrich, Korea). Anticoagulant buffer (ACB, pH 4.5) was prepared to contain 186 mM NaCl, 17 mM Na_2_EDTA, and 41 mM citric acid. Phosphate-buffered saline (PBS) was prepared to contain 100 mM phosphate at pH 7.4. Fluorescein isocyanate (FITC) labeling *Escherichia coli* was prepared with the method described by Shrestha and Kim [25].

### 2.3. Fractionation of Bacterial Culture Broth 

A culture broth of *X. hominickii* was centrifuged at 12,500× *g* for 20 min at 4 °C to separate bacterial cells from the culture broth. The supernatant was used for subsequent fractionation (Figure 1). At the first step, the same volume (1 L) of hexane was mixed with the supernatant to obtain organic and aqueous phases. The aqueous phase was combined with the same volume of ethyl acetate. The same process was sequentially repeated with chloroform and butanol organic solvents to obtain organic extracts. These organic extracts were then dried with a rotary evaporator (N-1110 Eyela, Tokyo Rikakikai, Tokyo, Japan) at 20–40 °C. Resulting dried pellets containing metabolites were resuspended with 5 mL of methanol each. Metabolites dissolved in methanol were used to perform nodulation and PLA_2_ enzyme assays. The butanol extract was subjected to a chromatography column filled with silica gel 60 (0.063–0.200 mm; Merck, Darmstadt, Germany) using a gradient chloroform/methanol mixture with increasing amount of methanol from 100:0 to 0:100 (*v*/*v*). Each resulting subfraction was used for nodulation and PLA_2_ enzyme assays. Active subfractions were separated by a preparatory thin-layer chromatography (Merck) with chloroform: methanol: acetic acid (7.5:2:0.5, *v*/*v*).

### 2.4. Nodulation Assay 

To assess nodule formation, immune challenge was performed by injecting 3 µL of *Escherichia coli* Top10 (10^4^ cells/larva) (Invitrogen, Madison, WI, USA) through the abdominal proleg of L5 larvae using a microsyringe (Hamilton, Reno, NV, USA). After incubating at 25 °C for 8 h, test insects were dissected to count the number of melanized nodules under a stereo microscope (Stemi SV11, Zeiss, Jena, Germany) at 50× magnification. For chemical test, 1 µL aliquot of test chemical at different concentrations was injected into each larva along with *E. coli*. AA was injected at 10 μg per larva. Each treatment was independently replicated five times.

### 2.5. Phagocytosis Assay

For in vivo phagocytosis assay, test chemical (1 µL) and FITC-labeled *E. coli* (5 µL, 5 × 10^4^ cells/µL) were injected to L5 larva. Hemolymph (50 µL) was collected at 15 min after injection and mixed with 50 µL of ice cold ACB. After centrifugation at 700 × *g* for 3 min at 4 °C, the supernatant plasma was replaced with 100 µL of TC-100 insect cell culture medium (Welgene, Daegu, Korea). Hemocyte suspension (10 µL) was placed onto a slide and over laid with 10 µL glycerol (50%, *v*/*v*). Hemocytes undergoing phagocytosis were determined by fluorescence-emitting cells under a fluorescence microscope (Leica, Wetzlar, Germany) at 40× magnification. For in vitro phagocytosis, hemocyte suspension and labeled bacteria mixture (30 µL) were incubated in a dark place for 1 h at room temperature. Hemocytes were then washed three times with PBS by centrifugation at 700× *g* for 2 min at 4 °C and finally resuspended in TC-100 medium. This hemocyte suspension was observed under the fluorescence microscope in FITC or DIC mode after forming monolayers on glass slides. 

### 2.6. Measurement of Nonspecific PLA_2_ Activity

Hemocyte PLA_2_ activity was fluorometrically determined with a pyrene-labeled phospholipid substrate using a spectrofluorometer (VICTOR multi-label plate reader, PerkinElmer, Waltham, MA, USA) at excitation and emission wavelengths of 345 and 398 nm, respectively. For measuring PLA_2_ activity, hemolymph was collected from ~10 L5 larvae. Hemocytes and plasma were separated by centrifugation at 800× *g* for 3 min at 4 °C. Hemocytes were then homogenized in 500 µL of PBS with a polytron (Ultra-Turrax T8, Ika Laboratory, Funkentshort, Germany). After centrifugation at 14,000× *g* for 5 min at 4 °C, protein concentration of the supernatant was determined by Bradford [26] assay using bovine serum albumin (BSA) as the standard. The reaction mixture was prepared in a 96-well microplate by adding 129 μL of 50 mM Tris buffer (pH 7.0), 2 μL of 1 M CaCl_2_, and 2 μL of 10% BSA into each well. A sample solution consisted of 5 μL of test bacterial metabolites and 10 μL of hemocyte enzyme extract (125.2 μg/μL). The plate was then incubated at room temperature for 30 min. Enzyme reaction was then initiated by adding pyrene-labeled PL substrate (2 μL) to the reaction mixture. 

### 2.7. Characterization of PLA_2_ Inhibition

To characterize PLA_2_ inhibition by test chemicals, activities of two different PLA_2_s [secretory PLA_2_ (sPLA_2_) and cytosolic PLA_2_ (cPLA_2_)] were assessed with methods described by Vatanparast et al. [27].

### 2.8. Thin Layer Chromatography (TLC) Analysis of Bacterial Metabolites 

TLC was performed to analyze extracts of bacterial culture broth with organic solvents. Each organic extract was spotted at the bottom of a silica gel plate (20 × 20 cm; Merck) and then placed in a shallow pool of a mixture of chloroform, methanol, and acetic acid (7:2.5:0.5, *v*/*v*) as an eluent in a development chamber. The solvent was then allowed to run by capillary action until it reached to the top end of the plate. The silica gel plate was then removed and dried. Separated components were stained with a mixture (19:1, g/g) of sea sand (Merck) and iodine (Duksan, Ansan, Korea). To separate compounds, a preparatory TLC (20 × 20 cm, Merck) was performed using the same eluent. Target spots were scratched and dissolved in methanol. 

### 2.9. Monitoring Metabolite Fractionation with High-Performance Liquid Chromatography (HPLC) 

Purified samples were analyzed by HPLC. Samples fractionated with organic solvents were cleaned with a disposable membrane filter (pore size: 0.45 μm, Dismic-25HP, Advantec, Tokyo, Japan). The cleaned sample (20 μL) was injected into an HPLC (Agilent Technologies 1200 series, Santa Clara, CA, USA) equipped with a C18 column (Zorbax 300SB, Agilent Technologies). The sample was then separated with a mobile phase of acetonitrile-formic acid (0.2% formic acid in water and 0.2% formic acid in 80% acetonitrile) at a flow rate of 1 mL/min for 75 min and detected with UV at 280 nm with a temperature of 30 °C.

### 2.10. Chemical Determination Using Gas Chromatography-Mass Spectrometry (GC-MS) 

GC-MS analysis was done using a GC (7890B, Agilent Technologies) coupled with an MS (5977A Network, Agilent Technologies). The GC had an HP 5 MS column (nonpolar column, Agilent Technologies) with an internal diameter of 30 m × 250 μm and a film thickness of 0.25 μm. The carrier gas was helium at a flow rate of 1 mL/min. The injector temperature was 200 °C. The injection mode was set at a split mode with a split ratio of 10:1. The initial oven temperature was set at 100 °C, held for 3 min, and then raised to 300 °C at a rate of 5 °C/min. The final temperature of 300 °C was kept for 10 min. The total run-time was 53 min. Mass spectra were recorded in EI mode at 70 eV with scanning range of 33–550 m/z. Purified samples were respectively identified by comparing mass spectra of compounds with those deposited in the database (NIST 11, Version 2.0, NIST, Gaithersburg, MD, USA).

### 2.11. Cytotoxicity Test

Cytotoxicity of each secondary metabolite was evaluated by 3-(4,5 dimethylthiazol-2-yl)-2,5-diphenyl tetrazolium bromide (MTT) assay according to the method described by Mollah et al. [28]. with a slight modification. Briefly, Sf9 cells (1.2 × 10^4^ cells in 200 μL of TC-100 medium) were seeded into each 96-well. After adding each test chemical (2.5 μL in DMSO), the plate was incubated at 28 °C for 24 h. Control cells were treated with 2.5 μL of DMSO. After 24 h incubation, 10 μL of MTT (5 mg/mL in PBS) was added to each well and cells were cultured for another 4 h at 28 °C. Formazan granules formed in viable cells were dissolved in 50 μL of DMSO. Absorbance was then measured at 540 nm using a microplate reader (Victor Multilabel Plate Reader, PerkinElmer, Waltham, MA, USA). 

### 2.12. Effect of Secondary Metabolites on Pathogenicity of Bacterial Pathogens

For the virulence test of *X. hominickii*, 24 h-cultured cells (10^8^ cfu/mL) were diluted with sterile PBS. After surface sterilization with 95% ethanol, each L5 larva was injected with 1 µL of test compound (1 µg/larva) along with bacterial suspension (10^2^ cfu/larva). Injection was performed through the first abdominal proleg using a micro syringe. Injected larvae were incubated at room temperature individually in 9 cm Petri dishes. For oral toxicity assay of *B. thuringiensis*, a small piece (2 × 2 cm) of Chinese cabbage was dipped in 500 ppm of Bt or a mixture of Bt (500 ppm) with 500 ppm of test chemical for 5 min. After a brief (5 min) drying under a dark condition, the treated leaf was supplied to individual test larva in 9 cm Petri dish for 24 h at room temperature. After 24 h, larvae were fed with fresh cabbage leaves. Each treatment was replicated three times with 10 larvae per replication. Mortality was observed at four days after treatment. Larvae were considered dead if they were unable to move in a coordinated manner when prodded with a blunt probe.

### 2.13. Statistical Analysis

All data for continuous variables were subjected to one-way analysis of variance (ANOVA) using PROG GLM in SAS program [29]. Mortality data were subjected to arcsine transformation and used for ANOVA. Means were compared with the least significant difference (LSD) test at Type I error of 0.05. Median toxicity values such as median lethal dose (LD_50_), median lethal concentration (LC_50_), median lethal time (LT_50_), and median inhibition concentrations (IC_50_) were calculated by Probit analysis using EPA Probit Analysis Program, ver. 1.5 (U.S. Environmental Protection Agency, Washington, DC, USA).

## 3. Results

### 3.1. Fractionation of Bacterial Culture Broth and Inhibitory Activities against S. exigua Hemocyte PLA_2_

Bacterial culture broth (1 L) of *X. hominickii* was fractionated with four different organic solvents (Figure 1). Resulting extracts were 0.05 mg of hexane extract, 1.0 mg of ethyl acetate extract, 0.50 mg of chloroform extract, and 2.0 mg of butanol extract. Each extract was dissolved in 0.4 mg/mL of methanol. Their compositions were analyzed by HPLC (Appendix A). Some early peaks before 10 min retention time appeared to be overlapped among these four extracts. However, later peaks were not apparently overlapped among the four extracts. In particular, butanol extract had more peaks than the other three extracts.

These four organic extracts were assessed for their inhibitory activities against hemocyte PLA_2_ of *S. exigua* (Figure 2A). All four extracts significantly (*p* < 0.05) inhibited PLA_2_ activities, with butanol extract exhibiting the most potent inhibitory activity. These four extracts also inhibited cellular immune responses assessed by nodule formation (Figure 2B). Such suppressed immune response was significantly (*p* < 0.05) rescued by AA addition.

### 3.2. Subfractionation of Butanol Extracts Identifies Eight PLA_2_ Inhibitor Candidates

Butanol extract was found to be the most potent one for inhibiting PLA_2_ activity. It contained diverse compounds (Appendix A). Therefore, it was further fractionated by silica gel chromatography under a gradient increase of methanol mixed with chloroform, resulting in 15 subfractions (F1-F15, Figure 1). These 15 fractions were assessed for their inhibitory activities against hemocyte PLA_2_ of *S. exigua* (Appendix A). Two fractions (F2 and F6) exhibited significantly (*p* < 0.05) higher inhibitory activities against PLA_2_ than other fractions. These two subfractions contained different compounds based on HPLC analysis (Appendix A). They were separated by preparatory TLC (Appendix A). Each TLC band was separately collected and used for subsequent PLA_2_ assay. A total of 18 subfractions were obtained from F2 and F6. They were adjusted to a concentration of 1 µg/µL in methanol. When these subfractions were assessed for their inhibitory activities against hemocyte PLA_2_ of *S. exigua*, eight subfractions exhibited significant (*p* < 0.05) higher inhibitory activities against PLA_2_ (Figure 3A). These active fractions were also assessed for their inhibitory activities against cellular immune responses (Figure 3B). All eight subfractions significantly (*p* < 0.05) inhibited nodule formation. Such suppressed immune responses were rescued by AA addition. When these eight active fractions were assessed by HPLC, they all contained more than one compound (Appendix A). However, most of them had a main peak. Thus, these eight fractions were subjected to GC-MS analysis (Appendix A). Eight different compounds were predicted from their main peaks: dioctyl terephthalate (DOTP) from F2-1, 3-ethoxy-4-methoxyphenol (EMP) from F2-2, bis(2-ethylhexyl) phthalate (BEP) from F2-6, 2-ethyl-1-hexanol (EH) from F2-8, docosane (DS) from F6-3, phthalimide (PM) from F6-4, o-cyanobenzoic acid (CBA) from F6-8, and dibutylamine (DBA) from F6-9. 

### 3.3. PLA_2_-Inhibitory Activities of Eight Compounds Identified from X. hominickii Culture Broth

Eight candidates identified were tested for their inhibitory activities against PLA_2_ using their synthetic compounds. All eight compounds significantly inhibited the activity of hemocyte PLA_2_ of *S. exigua* (Figure 4A). Among them, EMP and PM were the most potent ones while CBA was the least potent one based on their median inhibitory concentrations (IC_50_) (Table 1). To clarify whether their specific target was PLA_2_, two different PLA_2_ (sPLA_2_ and cPLA_2_) were assessed for their susceptibilities to these eight compounds (Figure 4B,C). Based on IC_50_ values, BEP, EH, and EMP were highly potent for inhibiting the activity of sPLA_2_ while CBA and DS were the least potent ones (Table 1). For inhibiting the activity of cPLA_2_, DOTP, EMP, and PM were highly potent while CBA was the least potent one. 

### 3.4. Immunosuppressive Activities of Eight Compounds Identified from X. hominickii Culture Broth 

Hemocytes of *S. exigua* could perform phagocytosis to remove bacteria (Figure 5A). Treatment with each of the eight compounds suppressed this cellular immune response (Figure 5B). Based on IC_50_ values, EMP and PM were highly potent in inhibiting phagocytosis while CBA was the least potent one (Table 2). Such phagocytosis-inhibiting activities of these eight compounds were all reversed by AA addition (Figure 5C). 

Nodulation is another cellular immune response of *S. exigua* against bacterial pathogens (Figure 6A). EMP and PM treatments exhibited high potency in inhibiting this cellular immune response (Table 2). Such inhibited cellular immune responses of all compounds were rescued by AA addition (Figure 6B).

### 3.5. Insecticidal Activities of Eight Compounds Identified from X. hominickii Culture Broth

Several PLA_2_ inhibitors isolated from entomopathogenic bacteria have been reported to exhibit cytotoxicity and insecticidal activities [28]. Thus, we tested cytotoxic effects of the eight compounds identified from *X. hominickii* on Sf9 cells (Table 3). EMP and PM were the most potent ones in inducing cytotoxicities at concentration less than 1 µg/mL. When these eight compounds were individually injected to *S. exigua* larvae, all compounds killed the larvae in a dose-dependent manner. Among these compounds, EMP and PM were the most potent ones for killing the larvae (Table 3). These compounds were also tested for their feeding toxicities using a leaf-dipping method. They showed significant (*p* < 0.05) oral toxicities without a feeding avoidance behavior. Among these compounds, EMP and PM were the most potent for killing the larvae. 

The addition of each of these eight compounds to *X. hominickii* significantly increased the pathogenicity of those bacteria (Appendix A). Such increase was more noticeable with the addition of EMP or PM. These two compounds were further analyzed for their abilities to enhance bacterial pathogenicity. The addition of either EMP or PM to two bacterial pathogens of *X. hominickii* (Figure 7A) and *B. thuringiensis* (Figure 7B) significantly (*p* < 0.05) increased their insecticidal activities. 

## 4. Discussion

A mutualistic interaction between nematodes and bacteria has been well-demonstrated in *S. monticolum-X. hominickii* system [22]. In the nematode-bacterial symbiotic system, host nematodes will deliver symbiotic bacteria to target insects’ hemocoels, in which bacteria will performs a crucial mutualistic activity by suppressing the immune responses of target insects to provide a protected environment for host nematodes. A previous study [23] has shown that *X. hominickii* can suppress insect immune responses by inhibiting PLA_2_ enzyme activity using secondary metabolites including oxindole. These findings suggest that *X. hominickii* might produce other PLA_2_ inhibitors. The current study identified eight different PLA_2_ inhibitors from the culture broth of *X. hominickii*.

To search for novel PLA_2_ inhibitors, this study analyzed butanol extract of the culture broth of *X. hominickii*. Among four organic extracts, the butanol extract showed significantly higher inhibitory activities against PLA_2_ of *S. exigua* hemocytes. Indeed, the butanol extract possessed more diverse compounds than other organic extracts based on HPLC analysis. Further fractionation of metabolites using chloroform-methanol showed that PLA_2_ inhibitors were eluted in hydrophobic fractions from the butanol extract. Final purifications with a preparative TLC coupled with functional assays identified eight different PLA_2_ compounds. 

These PLA_2_ compounds identified in this study are known in other systems with biological activities other than PLA_2_ inhibition. Phthalimide (PM) has a low acute mammalian toxicity with LD_50_ > 5000 mg/kg after oral administration [30]. PM derivatives have interesting activities, including antibacterial, anti-inflammatory, anticonvulsant, antiviral, and antitumor activities [31]. They have also been applied in agriculture as herbicides, fungicides, and insecticides [32,33]. Dioctyl terephthalate (DOTP) is relatively nontoxic to mammals, although there is a toxicity report about its effect on reproduction when its dose is too high [34]. Docosane (DS) is a straight-chain alkane with 22 carbon atoms and a water solubility at 7.77 × 10^−7^ mg/L at 25 °C [35]. A homologous series of n-alkanes ranging from n-C12~n-C31 have been found in all samples of bovine tissues, indicating that they have little toxicities. DS can also originate from some plants. There are several reports on the chemical composition and antibacterial activity of *Origanum vulgare* and *O. acutidens* against various bacteria of food, clinical, and plant origins [36,37,38]. They also have insecticidal effects [39]. Indeed, a hexane extract of an aromatic plant, *O. vulgare*, has a potential antibacterial effect with n-Docosane along with other constituents [40]. 2-Ethyl-1-hexanol (2-EH) is a branched, eight-carbon chiral alcohol. It is present in natural plant fragrances. Its S and R isomers have different odors [41]. It exhibits a low toxicity in animal models, with LD_50_ ranging from 2 to 3 g/kg in rats [42]. Bis-(2-ethylhexyl)-phthalate (BEP) is the most common member of the class of phthalates. Its acute toxicity is low in animal models (30 g/kg in rats by oral administration and 24 g/kg in rabbits by dermal application). However, it is potentially an endocrine disruptor [30]. BEP has a cytotoxic effect [43]. Qiao et al. [44] have observed that 4-cyanobenzoic acid (CBA) can inhibit activities of both monophenolase and diphenolase of mushroom tyrosinase, with IC_50_ at 2.45 mM for monophenolase and 1.40 mM for diphenolase. Zanjani et al. [45] have found that 3-ethoxy-4-methoxyphenol (EMP) is a constituent of essential oil of *Mentha pulegium* with an antimicrobial activity. Gram-positive bacteria like *Staphylococcus aureus* and *Bacillus cereus* are more sensitive to this essential oil than Gram-negative *E. coli* bacteria [46,47]. Dibutylamine (DBA) has an acute oral toxicity (LD_50_) at 1.89 g/kg in rats and an acute dermal toxicity (LD_50_) at 7.68 g/kg in rabbits. It also exhibits an antibacterial activity [48]. These eight PLA_2_ inhibitors are different in skeleton structures (phenylethyl or indole derivatives) known in previous PLA_2_ inhibitors identified from ethyl acetate extracts of *Xenorhabdus* and *Photorhabdus* [20]. Oxindole is another PLA_2_ inhibitor identified from ethyl acetate extract of *X. hominickii*. Thus, a total of nine PLA_2_ inhibitors were identified from *X. hominickii*. 

There were significant variations among the eight identified PLA_2_ inhibitors in this study in their enzyme-inhibiting activities and subsequent immunosuppression effects on target insects. Compared to six other compounds, two structurally different EMP and PM had higher inhibitory activities against hemocyte PLA_2_, in which EMP was highly inhibitory to both sPLA_2_ and cPLA_2_ activities while PM was highly inhibitory to cPLA_2_ activity. Catalytic activity of PLA_2_ is required for the biosynthesis of eicosanoids [49]. Eicosanoids play a crucial role in insect immune responses by activating PO [18] or by stimulating migration of hemocytes to infection foci [50]. Thus, inhibition of PLA_2_ can prevent de novo supply of eicosanoids in response to pathogen infection, leading to substantial enhancement of susceptibility of target insects to microbial pathogens. This prediction was supported by results of immunological assays using phagocytosis and nodule formation assays after bacterial challenge. Results showed that EMP and PM possessed significantly higher inhibitory effects compared to six other PLA_2_ inhibitors.

These eight PLA_2_ inhibitors exhibited cytotoxicities and insecticidal activities after oral administration. In particular, EMP and PM, which were highly inhibitory against hemocyte PLA_2_, exhibited the most potent cytotoxicity and insecticidal activity. Mollah et al. [28] have shown that PLA_2_ inhibitors are cytotoxic by inducing apoptosis with insecticidal activities. These results suggest that PLA_2_ inhibitors can induce cytotoxicity of insect cells, leading to fatal toxicity. However, it remains unclear how these PLA_2_ inhibitors can induce apoptosis. FAD-glucose dehydrogenase (GLD) is a prime signal of apoptosis induction of *S. exigua* hemocytes in response to infection by *X. nematophila* [51]. GLD is associated with the production of reactive oxygen species, a main inducer of apoptosis [52]. On the other hand, GLD plays a crucial role in mediating cellular immune responses in insects [53]. GLD is present in plasmatocytes and granules of granulocytes. However, its activity is only detected in immune-activated plasmatocytes [53]. GLD can catalyze the oxidation of glucose to gluconolactone. It participates in immune reactions by donating electrons during regeneration of its cofactor FAD. These electrons can be transferred to quinines as electron acceptors, resulting in superoxide anion radicals that are highly reactive [53] and required for hemocyte encapsulation [54]. Thus, GLD activity is required for activating cellular immune response of hemocytes before undergoing apoptosis. In this regard, PLA_2_ inhibitors released from *X. hominickii* might be able to activate GLD to induce inappropriate apoptosis of hemocytes and other cells, leading to fatal cytotoxicity. This is supported by an observation in a human cancer cell line (“U937”), in which BZA can induce apoptosis of by activating caspase-3 [55]. However, a direct functional link between BZA and GLD remains unknown. 

Immunosuppressive activities of these eight PLA_2_ inhibitors significantly increased insecticidal activities of two entomopathogens: *X. hominickii* and *B. thuringiensis* (Bt). Especially, the insecticidal activity of Bt was significantly enhanced by the addition of EMP or PM. This suggests that immune responses of insects play a crucial role in defending against Bt pathogenicity. Different immune-associated genes including antimicrobial peptides, serine proteases, and specific immune-associated Repat family genes in *S. exigua* are upregulated upon Bt infection [56]. Immunosuppression of *S. exigua* leads to enhanced Bt pathogenicity [57]. These results suggest that the eight PLA_2_ inhibitors identified in this study might have potential as novel insecticides.

## 5. Conclusions

This study identified eight different PLA_2_ inhibitors from *X. hominickii* culture broth. These PLA_2_ inhibitors suppress immune responses of target insects. They also showed cytotoxicities against insect cells and had potent insecticidal activities by injection or feeding treatments. With their immunosuppressive activities, they enhanced Bt pathogenicity. Especially, EMP and PM were the most potent.

## Figures and Tables

**Figure 1 insects-11-00505-f001:**
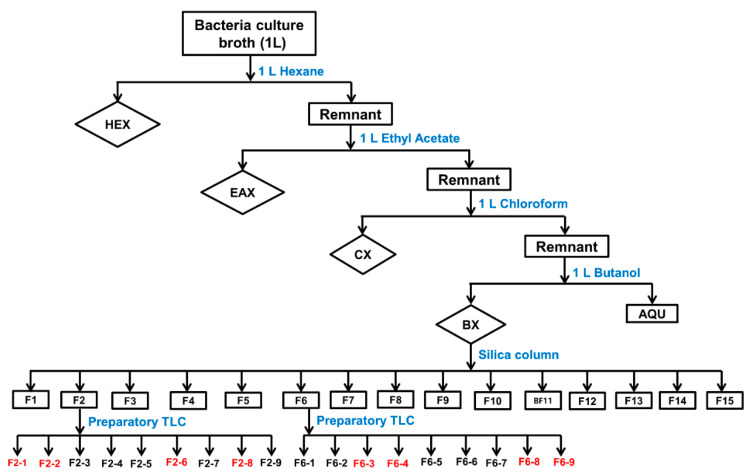
A diagram illustrating fractionation steps of culture broth of *Xenorhabdus hominickii*. Organic extracts were collected from hexane (HEX), ethyl acetate (EAX), chloroform (CX), and butanol (BX) organic solvents. The butanol extract was separated using a chromatography column filled with silica gel where a gradient chloroform/methanol mixture with increasing amount of methanol from 100:0 to 0:100 (*v*/*v*) was used. The active butanol fractions (F2 and F6) were further separated using preparatory thin-layer chromatography (TLC).

**Figure 2 insects-11-00505-f002:**
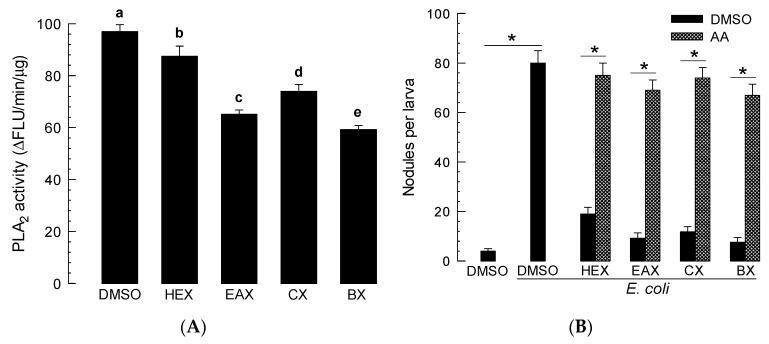
Screening four organic extracts of *X. hominickii* culture broth. Organic extracts were collected from hexane (HEX), ethyl acetate (EAX), chloroform (CX), and butanol (BX) organic solvents. (**A**) Inhibitory activities of the extracts against hemocyte phospholipase A_2_ (PLA_2_) of *S. exigua*. (**B**) Inhibitory activities of the extracts against hemocyte nodule formation. Each bacterial extract (1 µL/larva) were injected into L5 larvae along with *E. coli* (3 × 10^4^ cells/larva). Arachidonic acid (AA, 10 µg/larva) was injected along with the extract and *E. coli*. Dimethyl sulfoxide (DMSO) was used for solvent control. Each measurement was replicated three times with independent samplings. Different letters or asterisks above standard deviation bars indicate significant differences among means at Type I error = 0.05 (least significant difference (LSD) test).

**Figure 3 insects-11-00505-f003:**
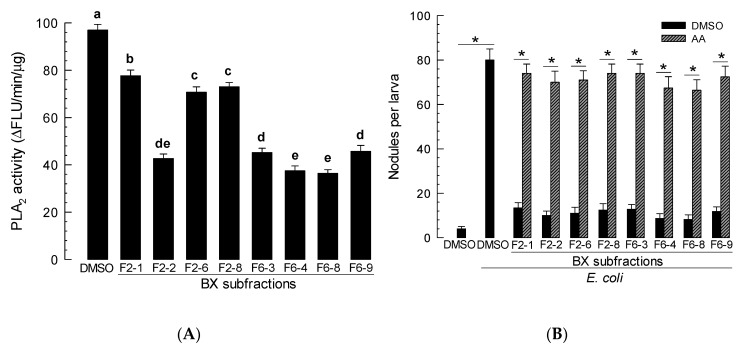
Screening eight butanol subfractions of *X. hominickii* culture broth. (**A**) Inhibitory activities of the subfractions against hemocyte PLA_2_ of *S. exigua*. (**B**) Inhibitory activities of the subfractions against hemocyte nodule formation. Each bacterial extract (1 µL/larva) were injected into L5 larvae along with *E. coli* (3 × 10^4^ cells/larva). Arachidonic acid (AA, 10 µg/larva) was injected along with the extract and *E. coli*. Dimethyl sulfoxide (DMSO) was used for solvent control. Each measurement was replicated three times with independent samplings. Different letters or asterisks above standard deviation bars indicate significant differences among means at Type I error = 0.05 (LSD test).

**Figure 4 insects-11-00505-f004:**
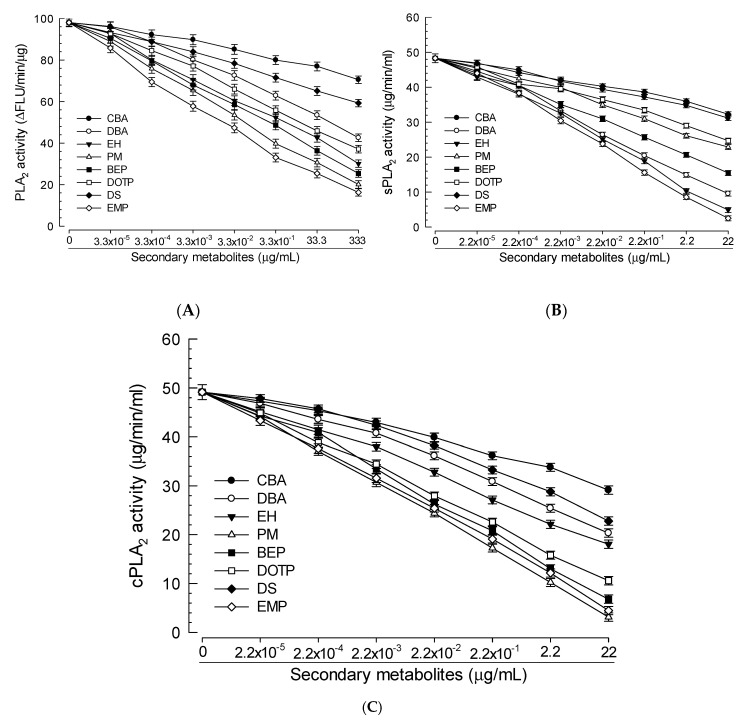
Inhibitory activity of bacterial secondary metabolites against three different PLA_2_ activities of *S. exigua*: nonspecific PLA_2_ (**A**), secretory PLA_2_ (sPLA_2_) (**B**), and cytosolic PLA_2_ (cPLA_2_) (**C**). The eight compounds are bis(2-ethylhexyl) phthalate (BEP), o-cyanobenzoic acid (CBA), dibutylamine (DBA), dioctyl terephthalate (DOTP), docosane (DS), 2-ethyl-1-hexanol (EH), 3-ethoxy-4-methoxyphenol (EMP), and phthalimide (PM). Each measurement was replicated three times with three independent samples.

**Figure 5 insects-11-00505-f005:**
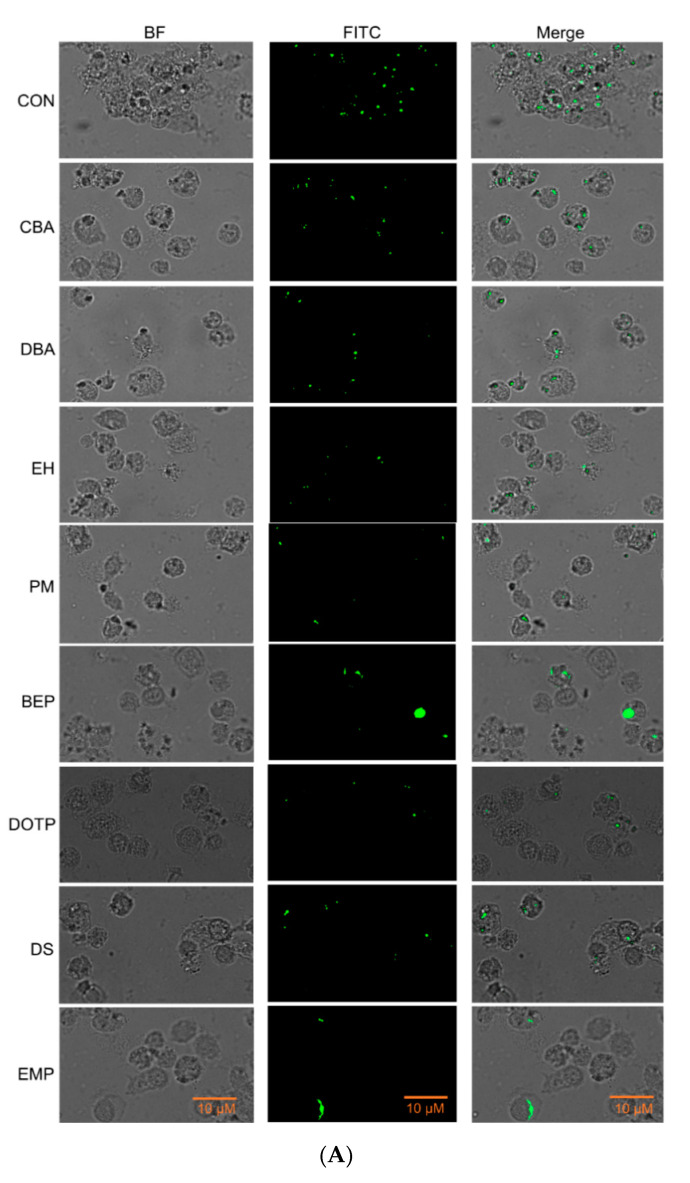
Immunosuppressive activities of eight PLA_2_ inhibitors against hemocyte phagocytosis of *S. exigua*. The eight compounds are bis(2-ethylhexyl) phthalate (BEP), o-cyanobenzoic acid (CBA), dibutylamine (DBA), dioctyl terephthalate (DOTP), docosane (DS), 2-ethyl-1-hexanol (EH), 3-ethoxy-4-methoxyphenol (EMP), and phthalimide (PM). (**A**) Inhibition of phagocytosis by the inhibitor treatments at 10 µg/larva. Phagocytic cells are recognized by green color emitted by FITC-labeled *E. coli*. Control (CON) used dimethyl sulfoxide (DMSO) for solvent control. (**B**) Dose effects of the inhibitors on suppressing phagocytosis. (**C**) Rescue effects of arachidonic acid (AA’, 10 µg/larva) on the phagocytosis suppressed by the inhibitors. Each treatment was replicated three times with independent insect samples. Asterisks above standard deviation bars indicate significant differences between treatments at Type I error = 0.05 (LSD test). ‘NS’ stands for no significant difference.

**Figure 6 insects-11-00505-f006:**
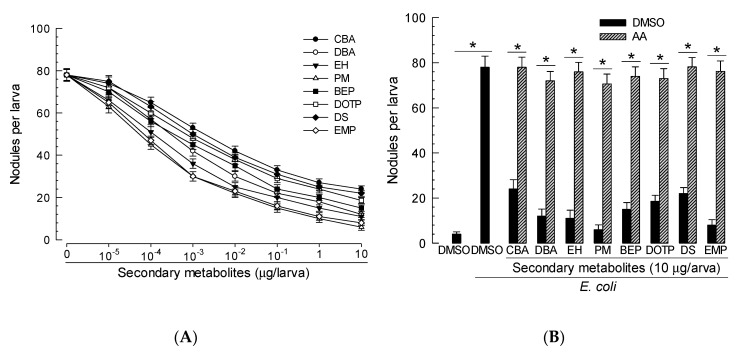
Immunosuppressive activities of eight PLA_2_ inhibitors against nodule formation of *S. exigua* against bacterial infection. The eight compounds are bis(2-ethylhexyl) phthalate (BEP), o-cyanobenzoic acid (CBA), dibutylamine (DBA), dioctyl terephthalate (DOTP), docosane (DS), 2-ethyl-1-hexanol (EH), 3-ethoxy-4-methoxyphenol (EMP), and phthalimide (PM). (**A**) Dose effects of the inhibitors on suppressing nodulation. (**B**) Rescue effects of arachidonic acid (AA’, 10 µg/larva) on the phagocytosis suppressed by the inhibitors. Control used dimethyl sulfoxide (DMSO) for solvent control. Each treatment was replicated three times with independent insect samples. Asterisks above standard deviation bars indicate significant differences between treatments at Type I error = 0.05 (LSD test).

**Figure 7 insects-11-00505-f007:**
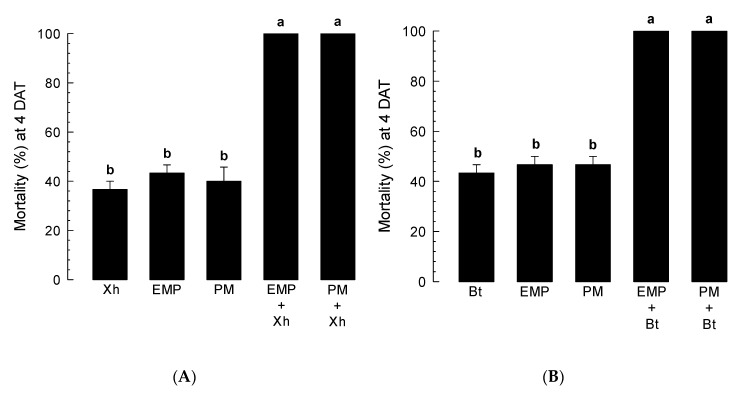
Enhanced insecticidal activities of bacterial pathogens by two potent PLA_2_ inhibitors: 3-ethoxy-4-methoxyphenol (EMP) and phthalimide (PM). (**A**) Enhanced effect of the inhibitors on *X. hominickii* (Xh) pathogenicity. L5 larvae of *S. exigua* were hemocoelically injected with low dose (10^2^ cfu/larva) of Xh or along with the PLA_2_ inhibitors (5 µg/larva). (**B**) Enhanced effect of the inhibitors on *Bacillus thuringiensis* (Bt) pathogenicity. A small piece (2 cm^2^) of cabbage leaf was dipped in Bt (500 ppm) or a mixture with PLA_2_ inhibitor (500 ppm). The treated leaves were provided to test *S. exigua* larvae. Each treatment was replicated three times with 10 insects per replication. Different letters above standard deviation bars represent significant differences among means at Type I error = 0.05 (LSD test). Insecticidal activity was assessed at four days after treatment (DAT).

**Table 1 insects-11-00505-t001:** PLA_2_-inhibitory activities of eight compounds identified from *X. hominickii* culture broth.

Compounds ^1^	Median Inhibitory Concentrations (IC_50_), µg/mL		
	Total PLA_2_	sPLA_2_	cPLA_2_
BEP	0.17 ± 0.01 cd ^2^	0.07 ± 0.01 ef	0.09 ± 0.01 d
CBA	0.86 ± 0.02 a	1.11 ± 0.11 a	0.38 ± 0.03 a
DBA	0.18 ± 0.02 c	0.11 ± 0.02 de	0.19 ± 0.02 c
DOTP	0.16 ± 0.006 cd	0.56 ± 0.05 c	0.06 ± 0.005 e
DS	0.81 ± 0.04 b	1.22 ± 0.15 a	0.32 ± 0.02 b
EH	0.14 ± 0.01 d	0.05 ± 0.004 f	0.11 ± 0.01 d
EMP	0.04 ± 0.02 e	0.03 ± 0.005 f	0.05 ± 0.004 ef
PM	0.05 ± 0.007 e	0.17 ± 0.01 d	0.04 ± 0.003 f

^1^ Bis(2-ethylhexyl) phthalate (BEP), o-cyanobenzoic acid (CBA), dibutylamine (DBA), dioctyl terephthalate (DOTP), docosane (DS), 2-ethyl-1-hexanol (EH), 3-ethoxy-4-methoxyphenol (EMP), and phthalimide (PM). ^2^ Different letters indicate significant difference among means in each column at Type I error = 0.05 (LSD test).

**Table 2 insects-11-00505-t002:** Immunosuppressive activities of eight compounds identified from *X. hominickii* culture broth.

Compounds ^1^	Median Inhibitory Concentration (IC_50_), ng/larva	
	Phagocytosis	Nodulation
BEP	1.50 ± 0.22 d ^2^	1.0 ± 0.11 c
CBA	32.50 ± 2.35 a	2.50 ± 0.22 a
DBA	5.0 ± 0.58 c	0.71 ± 0.05 d
DOTP	8.70 ± 0.89 b	1.10 ± 0.07 b
DS	8.40 ± 0.68 b	1.11 ± 0.09 b
EH	0.5 0 ± 0.03 e	0.40 ± 0.03 e
EMP	0.20 ± 0.03 g	0.20 ± 0.02 f
PM	0.40 ± 0.05f	0.20 ± 0.02 f

^1^ Bis(2-ethylhexyl) phthalate (BEP), o-cyanobenzoic acid (CBA), dibutylamine (DBA), dioctyl terephthalate (DOTP), docosane (DS), 2-ethyl-1-hexanol (EH), 3-ethoxy-4-methoxyphenol (EMP), and phthalimide (PM). ^2^ Different letters indicate significant difference among means in each column at Type I error = 0.05 (LSD test).

**Table 3 insects-11-00505-t003:** Insecticidal activities of eight compounds identified from *X. hominickii* culture broth.

Compounds ^1^	Cytotoxicity (MIC ^2^, µg/mL)	Insecticidal Activities	
		Injection (µg/larva)	Feeding (µg/mL)
BEP	0.30 ± 0.01 d ^3^	34.88 ± 2.08 bc	2679.58 ± 60.58 d
CBA	0.50 ± 0.02 c	54.96 ± 4.03 a	2850.65 ± 80.68 c
DBA	0.80 ± 0.03 b	38.77 ± 3.26 b	2969.46 ± 50.56 c
DOTP	0.50 ± 0.01 c	38.89 ± 3.21 b	3220.45 ± 90.91 b
DS	1.0 ± 0.03 a	49.54 ± 4.55 a	3554.0 ± 100.18 a
EH	0.20 ± 0.02 e	29.90 ± 3.05 c	2456.35 ± 50.88 e
EMP	0.1 ± 0.01 f	18.95 ± 2.61 d	2100.34 ± 40.19 f
PM	0.1 ± 0.01 f	22.11 ± 2.69 d	2150.66 ± 50.07 f

^1^ Bis(2-ethylhexyl) phthalate (BEP), o-cyanobenzoic acid (CBA), dibutylamine (DBA), dioctyl terephthalate (DOTP), docosane (DS), 2-ethyl-1-hexanol (EH), 3-ethoxy-4-methoxyphenol (EMP), and phthalimide (PM). ^2^ Minimum inhibitory concentration (MIC) ^3^ Different letters indicate significant difference among means in each column at Type I error = 0.05 (LSD test).

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
