# Peer review of "Immunosuppressive Activities of Novel PLA2 Inhibitors from Xenorhabdus hominickii, an Entomopathogenic Bacterium"

_insects, 2020, doi:10.3390/insects11080505_

Round 1
Reviewer 1 Report
The manuscript is well written and interesting to read.
Authors have generated well designed experiments to examine a testable hypothesis.
Results are logically presented and support the autors conclusions.
The authors demonstrated synergy between PM, EMP, and Bt. I found this to be most interesting and merrits further investigation.
I would suggest demonstrating the rescue effect with a non-PLA2 fatty acid, such as palmitic acid, to strengthen the connection to eicosanoids. This would demonstrate that not just any fatty acid will rescue the effect. However, in my view, the manuscript is solid as is. I only suggest this as a point of discussion.
Author Response
Comment #1-1: The manuscript is well written and interesting to read. Authors have generated well designed experiments to examine a testable hypothesis. Results are logically presented and support the autors conclusions. The authors demonstrated synergy between PM, EMP, and Bt. I found this to be most interesting and merrits further investigation. I would suggest demonstrating the rescue effect with a non-PLA2 fatty acid, such as palmitic acid, to strengthen the connection to eicosanoids. This would demonstrate that not just any fatty acid will rescue the effect. However, in my view, the manuscript is solid as is. I only suggest this as a point of discussion.
Response: We performed the additional assay. Based on this assay, we added the following findings to Results: “The rescue effect was specific to PLA2-associated fatty acid (AA) because palmitic acid, a non-PLA2-associated fatty acid, did not rescue the immunosuppression (data not shown).”

Reviewer 2 Report
Good paper! I can tell the authors preformed a lot of intensive experiments to identify these compounds and test their toxicity against the target insect. The introduction and discussion however hinder the impact of the work done and should be written better. Some of the methodology is also missing, unclear, or sparse which would prevent others from replicating your experiments. The sections also have small grammatical errors (the missing of plural/singular confusion) – I tried and commented on some of this initially but did not do this throughout the manuscript because it became too extensive.
32. The introduction is too short and poorly setups the rational for the experiment.
33-42. The framing of the opening paragraph is weak and misleading. I always marvel at how even though insects have an innate immune system it is surprisingly complicated and robust, capable of feats adaptive immune systems would envy (alternative splicing of immune genes in fruit flies that have differential activity against pathogens and transgenerational immunity come to mind).
35. Pathogen attack or challenge instead of infection - seems inappropriate since the immune system will engage the pathogen/parasite regardless if it leads to an infection.
41. The prophenoloxidase cascade - which I might add is a serine protease cascade to activate proPO to PO. You can easily reword the sentence to state this.
43 - 45. Nematodes seem absent although are mentioned in the next paragraph - since you mention parasitoid eggs on line 44 it should be insect parasites as opposed to pathogens.
46. The transition to eicosanoid biosynthesis by activating PLA2 in insects is choppy and might need to be further elaborated or better incorporated.
49. the entomopathogenic nematode genera
51. as the mouth
54. under these immunosuppressive conditions
60. to the nematode Steinernema
66 -69. reads like methods
72 - 80. Insect rearing and bacterial culture - this is very scant. The full rearing methods need to be described - the diet is not even mentioned (did you formulate it yourself, is it a commercially available one, can you cite another paper and then briefly mention the methods?). The bacterial culture should be a separate section.
77. - There was no real mention of BT specifically in the introduction. I thought these experiments would concern Xenorhabdus hominickii the entomopathogenic bacterium symbiotic to the nematode Steinernema monticolum. Could you be clear about why you are also using Bt earlier in the manuscript.
81. Chemicals - you don't need to say “was purchased”, you can place the manufacture in parenthesis along with the concentration at the end.
91. and 4 C
113. an immune challenge
146 - 148. briefly describe method
222. Can you remind and go more into detail about AA and its rescue effects?
340. The feeding method assay does not appear to be in the cytotoxicity section of the methods.
370 - 376. This part reads more as results than discussion of your findings.
377 – 406. The third paragraph of the discussion is more a laundry list of facts and the synthesis of previous findings of these compounds could be better stated.
Author Response
Comment #2-1: Good paper! I can tell the authors preformed a lot of intensive experiments to identify these compounds and test their toxicity against the target insect. The introduction and discussion however hinder the impact of the work done and should be written better. Some of the methodology is also missing, unclear, or sparse which would prevent others from replicating your experiments. The sections also have small grammatical errors (the missing of plural/singular confusion) – I tried and commented on some of this initially but did not do this throughout the manuscript because it became too extensive.
Response: The missing M&M is added. For example, details are added to the assays of sPLA2 and cPLA2 enzyme measurements though they are cited with previous studies. Text was polished with English Editing Company. Additionally, the comments raised by reviewer were much helpful to improve the text.
Comment #2-2: 32. The introduction is too short and poorly setups the rational for the experiment.
Response: Comments raised by reviewer are first reflected to improve Introduction. In addition, the objective is shortened by deleting experimental procedures. However, the additional objective is now added to introduce Bt and application of PLA2 inhibitors.
Comment #2-3: 33-42. The framing of the opening paragraph is weak and misleading. I always marvel at how even though insects have an innate immune system it is surprisingly complicated and robust, capable of feats adaptive immune systems would envy (alternative splicing of immune genes in fruit flies that have differential activity against pathogens and transgenerational immunity come to mind).
Response: The sentence is rephrased as follows: “Insect immunity is innate with highly efficient cellular and humoral responses”
Comment #2-4: 35. Pathogen attack or challenge instead of infection - seems inappropriate since the immune system will engage the pathogen/parasite regardless if it leads to an infection.
Response: Reworded as follows: “Upon pathogen attack, insects…”
Comment #2-5: 41. The prophenoloxidase cascade - which I might add is a serine protease cascade to activate proPO to PO. You can easily reword the sentence to state this.
Response: Reworded as follows: “as well as a serine protease cascade to activate prophenoloxidase (PPO) to PO [6]…”
Comment #2-6: 43 - 45. Nematodes seem absent although are mentioned in the next paragraph - since you mention parasitoid eggs on line 44 it should be insect parasites as opposed to pathogens.
Response: Reworded as follows: “including bacteria, fungi, viruses, and nematodes in insects [7].”
Comment #2-7: 46. The transition to eicosanoid biosynthesis by activating PLA2 in insects is choppy and might need to be further elaborated or better incorporated.
Response: Reworded as follows: “Non-self-recognition can stimulate eicosanoid biosynthesis by activating PLA2 in insects, though the recognition-to-PLA2 activation is not clearly understood in molecular level [8,9].”
Comment #2-8: 49. the entomopathogenic nematode genera
Response: Added
Comment #2-9: 51. as the mouth
Response: Added
Comment #2-10: 54. under these immunosuppressive conditions
Response: Added
Comment #2-11: 60. to the nematode Steinernema
Response: Added
Comment #2-12: 66 -69. reads like methods
Response: Deleted and rephrased as follows: “The objective of this study was to identify novel bacterial metabolites responsible for PLA2 inhibition from X. hominickii. Candidate PLA2 inhibitors were then analyzed for their inhibitory activities against cellular immune responses. Furthermore, the immunosuppressive effects of the PLA2 inhibitors were applied to enhance the pathogenicity of Bacillus thuringiensis to develop novel insecticides.”
Comment #2-12: 72 - 80. Insect rearing and bacterial culture - this is very scant. The full rearing methods need to be described - the diet is not even mentioned (did you formulate it yourself, is it a commercially available one, can you cite another paper and then briefly mention the methods?). The bacterial culture should be a separate section.
Response: The section is divided into insect rearing and bacterial culture. The citation number 24 describes the details of artificial diet. Bacterial isolation and identification are well described in citation number 22. All are descried in M&M.
Comment #2-13: 77. - There was no real mention of BT specifically in the introduction. I thought these experiments would concern Xenorhabdus hominickii the entomopathogenic bacterium symbiotic to the nematode Steinernema monticolum. Could you be clear about why you are also using Bt earlier in the manuscript.
Response: Bt is added to the end of Introduction as follows: “Furthermore, the immunosuppressive effects of the PLA2 inhibitors were applied to enhance the pathogenicity of Bacillus thuringiensis to develop novel insecticides.”
Comment #2-14: 81. Chemicals - you don't need to say “was purchased”, you can place the manufacture in parenthesis along with the concentration at the end.
Response: Reworded as follows: “Arachidonic acid (AA: 5,8,11,14-eicosatetraenoic acid, Sigma-Aldrich Korea, Seoul, Korea) was dissolved in dimethyl sulfoxide (DMSO). A PLA2 surrogate substrate (1-hexadecanoyl-2-(1-pyrenedecanoyl)-sn-glycerol-3-phosphatidylcholine, Molecular Probes, Eugene, OR, USA) was dissolved in high grade ethanol (Sigma-Aldrich Korea).”
Comment #2-15: 91. and 4 C
Response: Confirmed the Celcius notation
Comment #2-16: 113. an immune challenge
Response: Added
Comment #2-17: 146 - 148. briefly describe method
Response: Added the method as follows: “Briefly, enzyme activities were measured using sPLA2 and cPLA2 assay kits (Cayman Chemical, Ann Arbor, MI, USA) containing arachidonoyl thio-PC and diheptanoyl thio-PC synthetic substrates, respectively. Ellman's reagent [5,5′-dithio-bis-(2-nitrobenzoic acid), DTNB] was used to create 5-thio-2-nitrobenzoic acid, a colored product.”
Comment #2-18: 222. Can you remind and go more into detail about AA and its rescue effects?
Response: Reworded as follows: “Such suppressed immune response was significantly (P < 0.05) rescued by AA (a catalytic product of PLA2) addition.”
Comment #2-19: 340. The feeding method assay does not appear to be in the cytotoxicity section of the methods.
Response: In 2.13 session, it is described as follows: “For oral toxicity assay of B. thuringiensis, a small piece (2 × 2 cm) of Chinese cabbage was dipped in 500 ppm of Bt or a mixture of Bt (500 ppm) with 500 ppm of test chemical for 5 min. After a brief (5 min) drying under a dark condition, the treated leaf was supplied to individual test larva in 9 cm Petri dish for 24 h at room temperature. After 24 h, larvae were fed with fresh cabbage leaves. Each treatment was replicated three times with 10 larvae per replication. Mortality was observed at four days after treatment. Larvae were considered dead if they were unable to move in a coordinated manner when prodded with a blunt probe.”
Comment #2-20: 370 - 376. This part reads more as results than discussion of your findings.
Response: Reworded as follows: “Immunosuppressive activities of these eight PLA2 inhibitors significantly increased insecticidal activities of two entomopathogens: X. hominickii and B. thuringiensis (Bt). Especially, the insecticidal activity of Bt was significantly enhanced by the addition of EMP or PM. This suggests that immune responses of insects play a crucial role in defending against Bt pathogenicity. Indeed, different immune-associated genes including antimicrobial peptides, serine proteases, and specific immune-associated Repat family genes in S. exigua are up-regulated upon Bt infection [56]. Immunosuppression of S. exigua by EMP or PM treatment leads to enhanced Bt pathogenicity [57]. These results suggest that the eight PLA2 inhibitors identified in this study might have potential as novel insecticides.”
Comment #2-21: 377 – 406. The third paragraph of the discussion is more a laundry list of facts and the synthesis of previous findings of these compounds could be better stated.
Response: Shortened as follows: “Phthalimide (PM) has a low acute mammalian toxicity with LD50 > 5,000 mg/kg after oral administration [30]. PM derivatives are known to have antibacterial, anti-inflammatory, anticonvulsant, antiviral, and antitumor activities [31-33]. Dioctyl terephthalate (DOTP) is relatively non-toxic to mammals with some toxicity against reproduction [34]. Docosane (DS) is a straight-chain alkane with 22 carbon atoms and a water solubility at 7.77 10-7 mg/L at 25oC [35]. DS derivatives derived from Origanum vulgare and O. acutidens exhibit antibacterial and insecticidal activities [36-40]. 2-Ethyl-1-hexanol (2-EH) is a branched, eight-carbon chiral alcohol and exhibits a low toxicity in animal models with LD50 ranging from 2 to 3 g/kg in rats [41,42]. Bis-(2-ethylhexyl)-phthalate (BEP) is the most common member of the class of phthalates. Its acute toxicity is low in animals, but it is potentially an endocrine disruptor [30]. BEP has a cytotoxic effect [43]. 4-Cyanobenzoic acid (CBA) is known to inhibit activities of both monophenolase and diphenolase of mushroom tyrosinase [44]. Zanjani et al. [45] have found that 3-ethoxy-4-methoxyphenol (EMP) is a constituent of essential oil of Mentha pulegium with an antimicrobial activity. Gram-positive bacteria like Staphylococcus aureus and Bacillus cereus are more sensitive to this essential oil than Gram-negative E. coli bacteria [46,47]. Dibutylamine (DBA) exhibits an antibacterial activity with relatively low mammalian toxicity [48].”

Reviewer 3 Report
The authors studied the “Immunosuppressive Activities of Novel PLA2 Inhibitors from Xenorhabdus hominickii, an Entomopathogenic Bacterium”. Manuscript provided a strong background and methodology for doing the work. The experimental set up of this study appears to be well-designed and the data collected carefully. However, the authors did not clarify results obtained in some experiments. In general, manuscript is interesting and the study was well conducted, but must be corrected before possible publication. Methodological errors and text editing are required:
L.30: Keywords should be in alphabetic order. Also, keywords serve to widen the opportunity to be retrieved from a database. To put words that already are into title and abstracts makes KW not useful. Please choose terms that are neither in the title nor in abstract.
L.74: Detail the artificial diet composition and nutrients.
L.113: Sentence starting “Immune challenge…”
Ls.117-118: Detail the preparation and concentrations used.
L.121: Sentence starting “Test chemical…”
L.146: Sentence starting “Activities of two…”
Ls.216-220: Provide statistical (F value, Degree Freedom, and P-value) data
Ls.233-252: Again statistical data are needed for each result.
Ls.330-360: Confusing. Results-based in Probit analysis is missing. Please, provide statistical information (chi-square, LD50 or LC50, confidential limits, P-value, slope, intercept) based in Probit analysis
Author Response
Comment #3-1: The authors studied the “Immunosuppressive Activities of Novel PLA2 Inhibitors from Xenorhabdus hominickii, an Entomopathogenic Bacterium”. Manuscript provided a strong background and methodology for doing the work. The experimental set up of this study appears to be well-designed and the data collected carefully. However, the authors did not clarify results obtained in some experiments. In general, manuscript is interesting and the study was well conducted, but must be corrected before possible publication. Methodological errors and text editing are required:
Response: The missing M&M is added. For example, details are added to the assays of sPLA2 and cPLA2 enzyme measurements though they are cited with previous studies. Text was polished with English Editing Company. Additionally, the comments raised by reviewer were much helpful to improve the text.
Comment #3-2: L.30: Keywords should be in alphabetic order. Also, keywords serve to widen the opportunity to be retrieved from a database. To put words that already are into title and abstracts makes KW not useful. Please choose terms that are neither in the title nor in abstract.
Response: Reworded as follows: “Secondary metabolite; Eicosanoid; Xenorhabdus hominickii; Insecticide; Immunity”
Comment #3-3: L.74: Detail the artificial diet composition and nutrients.
Response: The M&M section is divided into insect rearing and bacterial culture. The citation number 24 describes the details of artificial diet.
Comment #3-4: L.113: Sentence starting “Immune challenge…”
Response: Corrected
Comment #3-5: Ls.117-118: Detail the preparation and concentrations used.
Response: Reworded as follows: “1 µL aliquot of test chemical at different concentrations (10-5 10 μg/larva)”
Comment #3-6: L.121: Sentence starting “Test chemical…”
Response: Corrected
Comment #3-7: L.146: Sentence starting “Activities of two…”
Response: Corrected
Comment #3-8: Ls.216-220: Provide statistical (F value, Degree Freedom, and P-value) data
Response: Added as follows: “All four extracts significantly (F = 105.65; df = 4,10; P < 0.0001) inhibited PLA2 activities, with butanol extract exhibiting the most potent inhibitory activity. These four extracts also significantly (F = 538.36; df = 5,24; P < 0.0001) inhibited cellular immune responses assessed by nodule formation (Figure 2B).”
Comment #3-9: Ls.233-252: Again statistical data are needed for each result.
Response: Added as follows: “When these subfractions were assessed for their inhibitory activities against hemocyte PLA2 of S. exigua, eight subfractions exhibited significant (F = 318.79; df = 8,18; P < 0.0001) higher inhibitory activities against PLA2 (Figure 3A). These active fractions were also assessed for their inhibitory activities against cellular immune responses (Figure 3B). All eight subfractions significantly (F = 355.10; df = 9,40; P < 0.0001) inhibited nodule formation.”
Comment #3-10: Ls.330-360: Confusing. Results-based in Probit analysis is missing. Please, provide statistical information (chi-square, LD50 or LC50, confidential limits, P-value, slope, intercept) based in Probit analysis
Response: It is provided with supplementary data. Stat analysis is added to Supplementary data
Reviewer 4 Report
Iman et al describe the discovery of PLA2 inhibitors in entomopathogenic bacteria. The authors explain how they isolate and assess the anti-melanisation activity of 8 compounds isolated from these bacteria.
I enjoyed reading the manuscript, it is well written and concise. The introduction was on point. The extended methods are well explained.
The results and analysis are valid. I would just include or mention the chromatography and compound isolation in the text. It seems to jump from screening fractions to 8 inhibitory compounds (line 262). A simple sentence would suffice. I am also not a fan of Table 2 and Table 3. I would suggest making a graph so information can easily be seen. Again, this is to the discretion of authors.
The discussion was very interesting and shows a thorough literature research on those 8 compounds. The conclusions of the paper fit the results obtained in the study.
In general I find the manuscript very good and the topic/findings are interesting.
Author Response
Comment #4-1: Iman et al describe the discovery of PLA2 inhibitors in entomopathogenic bacteria. The authors explain how they isolate and assess the anti-melanisation activity of 8 compounds isolated from these bacteria. I enjoyed reading the manuscript, it is well written and concise. The introduction was on point. The extended methods are well explained.
The results and analysis are valid. I would just include or mention the chromatography and compound isolation in the text. It seems to jump from screening fractions to 8 inhibitory compounds (line 262). A simple sentence would suffice. I am also not a fan of Table 2 and Table 3. I would suggest making a graph so information can easily be seen. Again, this is to the discretion of authors.The discussion was very interesting and shows a thorough literature research on those 8 compounds. The conclusions of the paper fit the results obtained in the study. In general I find the manuscript very good and the topic/findings are interesting.
Response: Line 262 is reworded as follows: “Eight candidates identified from the bacterial culture broth were tested for their inhibitory activities against PLA2 using their synthetic compounds.” Table 2 and 3 are not changed due to more than three parameters. But we provide additional statistical data such as confidence intervals, chi test, df, P values in separate supplementary data.

Round 2
Reviewer 2 Report
The manuscript should be proofed again for grammatical errors
Introduction and discussion - I still think the intro/discussion could be elaborated on. If these chemicals are being screened to improve Bt toxicity you might want to add in and talk about control failures with Bt or why compounds such as yours can have additive or synergistic effects on pesticides (plus mention cases where this has been done). In the discussion you might like to add how this could be applied to other biopesticides too such as Nuclear Polyhedrosis Viruses which also have been weaponized to combat lepidopterans.
44 "pathogens" should be parasites and pathogens - nematodes or parasites
71 although the reference for the diet is cited it is proper to still lightly discuss the rearing regime - please add if the larvae were reared communally or solitarily, if the larvae are moved once they begin to wander/pupate, the size of the cage the adults were placed in, and what substrate was used for the adults to oviposite eggs on.
Author Response
Comment #2-1: The manuscript should be proofed again for grammatical errors
Response: Although the corresponding author wrote the manuscript, he performed the proof-reading again the entire manuscript and corrected followings:
L14 immunosuppresive è immunosuppressive
L44 Non-self-recognition è Nonself recognition
L45 the recognition-to-PLA2 è the recognition-to-PLA2 activation
L65 from X. hominickii è from the bacterial culture broth of X. hominickii
L93 see Fig. 1 è Figure 1
L95-97 shorten sentence to avoid confusion as follows: “The aqueous phase was combined with the same volume of ethyl acetate. The same process was sequentially repeated with chloroform, and butanol organic solvents to obtain organic extracts.”
L122 to be clear, the bacterial concentration is added as follows: “FITC-labeled E. coli (5 µL, 5 x 104 cells/µL)”
L155 replace “20 by 20” with 20 x 20
L238-239 reworded “potent among four organic extracts in inhibiting PLA2 activity.”
L274,278, 279, 299, 338, 341, 344 “one” deleted
L401,402 word correction “ethyl acetate”
L402 eight è nine (including previous identified compound)
L404 added “in this study” to exclude the previous known compound
L421 csa è can
Comment #2-2: Introduction and discussion - I still think the intro/discussion could be elaborated on. If these chemicals are being screened to improve Bt toxicity you might want to add in and talk about control failures with Bt or why compounds such as yours can have additive or synergistic effects on pesticides (plus mention cases where this has been done). In the discussion you might like to add how this could be applied to other biopesticides too such as Nuclear Polyhedrosis Viruses which also have been weaponized to combat lepidopterans.
Response: This study aimed to identify PLA2 inhibitors from an entomopathogen, Xenorhabdus hominickii because the bacterial pathogenicity comes from host immunosuppression by inhibiting PLA2. Fractionation of bacterial culture broth allowed us to identify eight different PLA2 inhibitors. Thus these eight inhibitors might be responsible for the bacterial pathogenicity. This was demonstrated by adding these PLA2 inhibitors to low bacterial dose of X. hominickii by enhancing pathogenicity. The pathogenicity activities of the PLA2 inhibitors were supported by applying them to Bt. Thus, the enhancement of Bt pathogenicity is a kind of demonstration of the immunosuppressive activities of the PLA2 inhibitors. We may focus on the application of these compounds to develop novel insecticides in subsequent researches. This future direction is described in the last paragraph of the discussion.
Comment #2-3: 44 "pathogens" should be parasites and pathogens - nematodes or parasites
Response: to be clear, it is changed into ‘entomopathogens”
Comment #2-4: 71 although the reference for the diet is cited it is proper to still lightly discuss the rearing regime - please add if the larvae were reared communally or solitarily, if the larvae are moved once they begin to wander/pupate, the size of the cage the adults were placed in, and what substrate was used for the adults to oviposite eggs on.
Response: Some details pertaining to this study are briefly described as follow: “A group (100 ~ 200 individuals) of larvae were reared on diet at a rearing box (25´20´10 cm) until they became 3rd instar. After then they were individually reared in 9-cm diameter Petri dish until pupation. Under this conditions, larvae underwent five instars (L1-L5).”.

Reviewer 3 Report
The manuscript “Immunosuppressive Activities of Novel PLA2 Inhibitors from Xenorhabdus hominickii, an Entomopathogenic Bacterium” has been improved and all my questions were taken into account. I recommend the publication in “Insects”.
Author Response
Comment #3-1: The manuscript “Immunosuppressive Activities of Novel PLA2 Inhibitors from Xenorhabdus hominickii, an Entomopathogenic Bacterium” has been improved and all my questions were taken into account. I recommend the publication in “Insects”.
Response: Thank you for wonderful review comments, which much improved this manuscript.
